# Evolution of Rheological Behaviors of Styrene-Butadiene-Styrene/Crumb Rubber Composite Modified Bitumen after Different Long-Term Aging Processes

**DOI:** 10.3390/ma12152345

**Published:** 2019-07-24

**Authors:** Yangsheng Ye, Gang Xu, Liangwei Lou, Xianhua Chen, Degou Cai, Yuefeng Shi

**Affiliations:** 1Railway Engineering Research Institute, China Academy of Railway Sciences Corporation Limited, Beijing 100081, China; 2School of Transportation, Southeast University, Nanjing 211189, China

**Keywords:** high speed railway, SBS/CR modified asphalt, long-term aging, anti-aging

## Abstract

In this study, a new type of composite modified bitumen was developed by blending styrene-butadiene-styrene (SBS) and crumb rubber (CR) with a chemical method to satisfy the durability requirements of waterproofing material in the waterproofing layer of high-speed railway subgrade. A pressure-aging-vessel test for 20, 40 and 80 h were conducted to obtain bitumen samples in different long-term aging conditions. Multiple stress creep recovery (MSCR) tests, linear amplitude scanning tests and bending beam rheometer tests were conducted on three kinds of asphalt binders (SBS modified asphalt, CR modified asphalt and SBS/CR composite modified asphalt) after different long-term aging processes, including high temperature permanent deformation performance, resistance to low temperature thermal and fatigue crack. Meanwhile, aging sensitivities were compared by different rheological indices. Results showed that SBS/CR composite modified asphalt possessed the best properties before and after aging. The elastic property of CR in SBS/CR composite modified asphalt improved the ability to resist low temperature thermal and fatigue cracks at a range of low and middle temperatures. Simultaneously, the copolymer network of SBS and CR significantly improved the elastic response of the asphalt SBS/CR modified asphalt at a range of high temperatures. Furthermore, all test results indicated that the SBS/CR modified asphalt possesses the outstanding ability to anti-aging. SBS/CR is an ideal kind of asphalt to satisfy the demand of 60 years of service life in the subgrade of high speed railway.

## 1. Introduction

Waterproofing layers are essential to preventing surface water from infiltrating into a high-speed railway subgrade, which can ensure its stability and bearing capacity, and especially prevent subgrade frost in seasonally frozen regions [1,2,3,4]. The dense-graded asphalt concrete was used as a waterproofing material to substitute for fiber-reinforced concrete, and its requirements were proposed based on the practice achievements in the Beijing–Zhangjiakou high-speed railway test section. Meanwhile, theoretical analysis, finite element calculation and past engineering experience show low temperature thermal and fatigue cracking, permanent deformation and passive stretching near the expansion joint of the base are the main failure modes of the asphalt concrete waterproof sealing layer [5,6,7], therefore, the ability for low temperature thermal and fatigue crack reduction and better deformation recovery are a prerequisite to extending the service life. It must be noted that according to high-speed railway standards in China, the design life of waterproofing layers in the high-speed railway is 60 years, which is four times the service life of the freeway. As is known, aging has a dramatic impact on the aforementioned performance of the asphalt binder or asphalt mixture [8,9,10,11].

However, conventional base asphalt is an easily aged material. Meanwhile, the dense-graded asphalt concrete used as waterproofing layers requires distinguished resistance to low-temperature crack, fatigue crack and permanent deformation. One of the most effective methods to achieve a better engineering performance of asphalt binders and mixtures to extend service life is to modify asphalts by specialized refining practices, chemical reactions and additives [12,13]. For instance, styrene-butadiene-styrene (SBS) modifier can improve permanent deformation at the range of the high-temperature domain and anti-aging performance of asphalt binders [14]. Crumb Rubber modified bitumen (CRMB) at high additive content decreases non recoverable deformation and enhances the high elastic response [15,16]. Montmorillonite or wood lignin modified asphalt binders are able to potentially delay the aging process and enhance deformation and fatigue resistance [17]. In recent years, a large number of researchers have begun to attach attention to compound modified asphalt and tend to combine the advantages of different modifiers [18]. For instance, the addition of crumb rubber (CR) and SBS can make an obvious improvement to asphalt in temperature sensitivity and the viscoelastic response behavior [19]. Ageing indexes of conventional parameters are greatly lessoned in SBS modified bitumen (SBSMB) by using the carbon nano-tubes [11]. 

The SBS/CR composite modification method is most commonly utilized among them. Because it can not only combine the advantages of CR and SBS to achieve a more satisfactory performance, but it can also cut back on environmental pollution using scrap tires [20]. In a number of research studies that studied the best preparation technology of SBS/CR modified bitumen (SBS/CRMB), it has been found that SBS/CRMB could raise high-temperature stability and possess the best aging resistance as SBS and rubber powder modified bitumen [11,18,19]. The SBS/CRMB with SBS and rubber powder in the end obtained the best preparation process reported by Wang et al. [21]. Inorganic or organic powders such as rubber powder, nano-TiO2 and carbon black can be added in order to raise the anti-aging ability of bitumen. However, the SBS/CRMB with a high performance also faced the great challenges of performances deterioration, which was caused by aging of asphalt materials, especially for the thermal oxygen aging and led to the weak viscoelastic performance of asphalt binder [9,10].

The objectives of this study were to select an ideal bitumen of asphalt concrete waterproofing layers. Therefore, in this research a new type of SBS/CRMB with a chemical modification method was developed. Dynamic Shear Rheological (DSR) tests and a Bending Beam Rheometer (BBR) test were conducted on SBS/CRMB before and after the short-term aging of thin film oven test (TFOT) and long-term aging of a pressurized aging vessel (PAV) aging for 20 h, 40 h and 80 h. Then, the rheological properties and the anti-aging properties of SBS/CRMB were evaluated.

## 2. Materials and Testing Methodology

### 2.1. Materials and Preparation

The base binder used in this study was provided by SK Co., Ltd., Korea (SK-70, PG 64-22) and the properties are shown in Table 1. SBS and crumb rubber modifier were provided by Jiangsu Baoli International Investment Co., Ltd. (Wuxi, China). Rubber processing oil, which is rich in aromatic and saturates was utilized to make crumb rubber and SBS swell sufficiently in bitumen. The content of rubber processing oil was 4% by mass of base asphalt. Sulfur powder acted as the cross-linking agent and the amount was 0.2% in bitumen weight. The content of SBS and crumb rubber in the SBS/CR composite modified binder (SBS/CRMB) were 5.5% and 15% in base asphalt weight, respectively. The production process of SBS/CR-MB was as follows: Firstly, heat the base asphalt to 165 °C, and then 4% of rubber processing oil and 15% of CR powder (by the mass of base asphalt), which was devulcanization in the laboratory and was added to the base asphalt and then blended by a special double screw extruder. Subsequently, they were sheared by a shearing machine with a rotation speed of 5000 rpm for about 30 min at 180 °C. Secondly, 5.5% wt of SBS was added to the aforementioned mixture, then sheared at a speed of 5000 rpm at 180 °C and the shearing time was observed during the preparation. Thirdly, 0.2% wt sulfur powder was added slowly to the sample at 1000 rpm and blended for about 15 min. Finally, the prepared samples were developed in an oven for 30 min.

As mentioned earlier, there were two different types of bitumen which were considered as references for the comparison of SBS/CR composite modified bitumen, namely SBSMB with 5.5% SBS and CRMB with 15% crumb rubber (by the weight of base asphalt). It should be noted that controlling the impact of the preparation on the binder’s properties was consistent, the same conditions were utilized except for the modifier during the preparation.

### 2.2. Test Methods

#### 2.2.1. Aging Method

The asphalt samples were carried out using TFOT aging at 163 °C for 5 h according to ASTM D1754 to simulate the short-term aging of asphalt during the mixing, transportation and paving progress. Meanwhile, the different long-term aged bitumen samples were obtained by conducting TFOT (5 h, 163 °C), followed by a PAV test for 20, 40 and 80 h with the temperature at 100 °C and 2.1 MPa pressure. The purpose of the longer PAV aging times was simply to create a more highly-aged sample and was not aimed to correlate with any expected services life.

#### 2.2.2. Multiple Stress Creep Recovery (MSCR) Test

The MSCR test was conducted as per AASHTO T350 using a dynamic shear rheometer (DSR) device. The diameter of samples used in the testing was 25 mm, the thickness was 1 mm and the testing temperature was 70 °C. The MSCR test result was based on two replicates. The MSCR test consisted of 20 cycles, a 1 s creep period and a 9 s recovery period at a stress level of 0.1 kPa and this was followed by another 10 cycles of creep and recovery at 3.2 kPa according to AASHTO T350. Under two stress conditions (Jnr-diff) the nonrecoverable creep compliance was different and is presented in Equation (1).
(1)$$J_{\mathrm{nr}-diff}=\frac{\left(J_{nr3.2}-J_{nr0.1}\right)}{J_{\mathrm{nr}0.1}}\times 100\%$$where $J_{nr0.1}$, $J_{\mathrm{nr}3.2}$ is the unrecoverable creep compliance at 0.1 kPa and 3.2 kPa (kPa^−1^), respectively. $J_{\mathrm{nr}-diff}$ is the difference in nonrecoverable creep compliance at the two stress levels (%).

#### 2.2.3. Linear Amplitude Scanning (LAS) Test

The LAS test was carried out to characterize the fatigue properties of all bitumen samples under different aging conditions at 25 °C. According to AASHTO TP 101-12, the testing samples in the LAS test were prepared circular with a diameter of 8 mm and a height of 2 mm. Bitumen samples were tested in two stages based on AASHTO TP 101. In the first stage, the frequency sweep test with a strain level of 0.1% was performed with different frequencies (0.2 to 30 Hz and change as a testing table), which was used to obtain the undamaged material parameter (α). In the second stage, a strain sweep test with a strain change from 0.1 to 30% linearly increased at a constant frequency of 10 Hz. The asphalt binder’s damage property was analyzed in viscoelastic continuum damage (VECD) mechanics. Three replicates were tested in this paper in order to guarantee the validity of the testing results.

The damage accumulation in the sample in the LAS test was calculated using Equation (2)
(2)$$D\left(t\right)≅\overset{N}{\underset{i=1}{\sum }}{\left[\pi \gamma_{0}^{2}\left(C_{i-1}-C_{i}\right)\right]}^{\frac{\alpha }{1+\alpha }}{\left(t_{i}-t_{i-1}\right)}^{\frac{1}{1+\alpha }}$$
where $C_{\left(t\right)}=\frac{|G^{*}|\left(t\right)}{|G^{*}{|}_{initial}}$ = integrity parameter;$G^{*}$ = complex shear modulus, MPa;$\gamma_{{}_{0}}$ = applied strain, %;t = testing time, s;*α* = 1/m, where m is the slope of the best-fit straight line with log (storage modulus) in vertical axis and log (applied frequency) on the horizontal axis;

(3)$$C_{\left(t\right)}=C_{0}-C_{1}{\left(D\left(t\right)\right)}^{C_{2}}$$
where C_0_ = 1, the initial value of C;C_1_, C_2_ = curve-fit coefficients, then change the form as shown below: (4)$$\mathrm{lg}\left(C_{0}-C_{\left(t\right)}\right)\;=\;\mathrm{lg}\left(C_{1}\right)+C_{2}\cdot \mathrm{lg}\left(D\left(t\right)\right)$$

The value of D(t) at failure, $D_{{}_{f}}$ is defined as the D(t) which corresponds to the reduction in initial |G*| at the peak shear stress. The calculation is as follows:(5)$$D_{f}={\left(\frac{C_{0}-C_{atpeakstress}}{C_{1}}\right)}^{\frac{1}{C_{2}}}$$
where $C_{atpeakstress}$ = $C_{\left(t\right)}$ at peak stress.

The following parameters (A and B) for the binder fatigue performance model can now be calculated and recorded as follows:(6)$$A=\frac{f{\left(D_{f}\right)}^{k}}{k{\left(\pi C_{1}C_{2}\right)}^{\alpha }}$$
where*f* = loading frequency (10 Hz),*k* = 1 + (1 – *C*_2_)α, andB = 2α.

The binder fatigue performance parameter $N_{f}$ can now be calculated as follows:(7)$$N_{f}=A{\left(\gamma_{max}\right)}^{B}$$
where: $\gamma_{\max}$ = the maximum expected binder strain, percent.

Meanwhile, the integrity parameter C which acts as the material integrity level can be calculated from Equation (8).
(8)$$C=\frac{\left|G^{*}\right|\sin\delta_{t}}{\left|G^{*}\right|\sin\delta_{inital}}$$
where:$\left|G^{*}\right|\sin\delta_{t}$ = the quotient of damaged value of $\left|G^{*}\right|\sin\delta$;$\left|G^{*}\right|\sin\delta_{\mathrm{inital}}$ = the initial undamaged value of $\left|G^{*}\right|\sin\delta$.

#### 2.2.4. Bending Beam Rheometer Test

A BBR test was employed to characterize the low-temperature performance of SBS-MB, CR-MB and SBS/CR-MB before and after long-term aging according to ASTM D6648. The test temperature was −12, −18 and −24 °C and the average results of three replicates were used as the testing results. Using the interpolation method according to ASTM D7643-16 it is possible to determine the low service temperature (T_L_) of asphalt binders from the BBR test in multiple testing temperatures. The critical temperature T_L,s_ and (T_L,m_) corresponding to stiffness = 300 MPa and m value = 0.3 were obtained by the regression Equations (9) to (10), respectively. The low service temperature (T_L_) was defined in Equation (11).
(9)$$\log_{10}\left(s\right)=a_{1}+b_{1}T$$
(10)$$m=a_{2}+b_{2}T$$
(11)$$T_{L}=\max\left(T_{L,s},T_{L,m}\right)-10$$
where:$a_{1}$, $a_{2}$, $b_{1}$ and $b_{2}$ are the regression coefficients;T = the test temperature (°C);$T_{L,s}$ = the critical temperature when S = 300 MPa (°C);$T_{L,m}$ = the critical temperature when m = 0.3 (°C);$T_{L}$ = the low service temperature (°C).

## 3. Results and Discussion

### 3.1. Multiple Stress Creep Recovery Test Results

In the MSCR test, the major parameters used to identify the permanent deformation performance of the asphalt binder included non-recoverable compliance ($J_{\mathrm{nr}}$), stress sensitivity ($J_{\mathrm{nr}-diff}$) and percent recovery [22]. Table 2 shows the MSCR test final results for all bitumen binders. A detailed analysis of permanent deformation and deformation recovery performance of three kinds of modified binders before and after long-term aging is provided below.

#### 3.1.1. Analysis of Non-Recoverable Compliance at 3.2 kPa

Figure 1 displays the J_nr3.2_ values for three kinds of modified bitumen at different long-term aging conditions. At the same testing temperature (70 °C), the J_nr3.2_ values of CR modified bitumen were clearly larger than SBS and SBS/CR modified binders, indicating that the SBS modifier plays an important role in anti-permanent deformation. The addition of SBS and CR in SBS/CR modified bitumen further increased the rutting performance of binders due to the copolymer network as described in previous research. With the increased aging conditions the J_nr3.2_ values of all kinds of bitumen increased, which was contrary to that of base asphalt due to aging leading to a stiffening effect for the binders. This phenomenon might be caused by the degradation of polymers during the aging process. Moreover, the J_nr3.2_ values of SBS/CR modified binders increased slightly from the condition of virgin to the aging of PAV for 80 h, followed by SBS modified binders. These results illustrate that aging is detrimental to permanent deformation of the decomposition of the polymer caused by thermal oxygen aging. It must be noted that because of the poor performance of CR modified bitumen in the MSCR and LAS tests, the PAV aging test of CR modified bitumen for 40 h and 80 h was abandoned.

#### 3.1.2. Evolution of Stress Sensitivity of All Kinds of Asphalt during the Aging Process

Figure 2 shows the J_nr, diff_ values for three kinds of modified bitumen at different aging conditions. At the testing temperature of 70 °C, J_nr, diff_ values of all tested binders were below 75%. Unlike the J_nr3.2_ values of all tested binders, there was no significant difference found in J_nr,diff_ values. Three kinds of modified binders held a similar stress sensitivity. Meanwhile, Jnr, diff values of three modified bitumen presented different trends during the process of aging. Jnr, diff values of CR and SBS modified bitumen before the PAV aging for 40 h increased with the extent of aging in contrast to that of SBS/CR modified bitumen. After PAV aging for 20 h, the Jnr, diff value at PAV aging for 40 h of SBS modified bitumen decreased suddenly, then decreased in the next aging condition of PAV aging at 80 h. However, the J_nr, diff_ values of SBS/CR modified binders in different aging conditions was relatively stable. Lower J_nr, diff_ values may be attributed to the aging of bitumen, which increased the binders’ stiffness or a stronger cross-link network of polymers. Therefore, the J_nr, diff_ values became complex during the process of aging [8].

#### 3.1.3. Analysis of Percent Recovery at 3.2 kPa

Percent recovery is a significant parameter which influences the deformation recovery of binder in the MSCR test. Figure 3 displays the percent recovery of the tested binders with the stress level of 3.2 kPa at 70 °C. CR modified bitumen owned the lowest percent recovery less than 10%, the other two modified bitumen possessed a much higher percent recovery. It indicated that the CR does not contribute to recovery behaviour of SBS/CR modified bitumen. The percent recovery of three modified bitumen decreased with the extension of aging time. Moreover, the percent recovery of SBS modified binders dropped rapidly after the aging of TFOT, but the percent recovery of SBS/CR modified binders decreased smoothly. This phenomenon may be attributed to the destruction of a three-dimensional network of SBS modifier in SBS modified bitumen, which decomposed faster than that in SBS/CR modified bitumen where there was a presence of carbon black, which released from CR during the process of the SBS/CR modified bitumen’s preparation. The carbon black could protect the SBS molecule from oxidizing.

AASHTO M 332 put forward a method to detect the polymer in the bitumen, which is shown in Equation (12). Figure 4 shows the relationship between the percent recovery and the Jnr value at 3.2 kPa for three kinds of binders in different long-term aging conditions. Binders’ percent recovery above the polymer modification curve demonstrates a good elastomeric behaviour. As showed in Figure 4, the percent recovery of CR modified bitumen is below the standard line in all aging conditions. Nevertheless, the SBS/CR- modified bitumen is always above the standard line regardless of aging conditions. The SBS modified binders were divided into two parts, on the one hand, when the PAV aging time was less than 40 h, the SBS-modified bitumen was up to the standard line and then below the standard line with increasing aging time. It demonstrated that the degradation of SBS modifiers in SBS/CR- modified binders was less than that of SBS modified binders. The SBS/CR modified bitumen employed an outstanding ability to resist aging.

(12)$$R=\left\{ \begin{array}{l} 29.37{\left(J_{nr3.2}\right)}^{-0.2633},J_{nr3.2}\geq 0.1\\ 55,\;J_{nr3.2}<0.1 \end{array}\right.$$

### 3.2. Linear Amplitude Scanning Test Results

#### 3.2.1. Evolution of the Damage Intensity and Integrity Parameters of All Kinds of Asphalt during the Aging Process

Table 3 shows the LAS test fatigue damage parameter for CRMB, SBS MB and SBS/CR MB in different aging conditions. Figure 5 displays the relationship between damage intensity (D) and the integrity parameter (C) for all tested binders. A binder possesses better fatigue performance with lower values of C_1_ and C_2_ [23,24,25]. Table 3 shows that the value of great majority C_1_ parameter increased with increasing aging time. However, the C_2_ parameter was irregular with the aging time. Aged SBS/CR modified binders employed a lower C_1_ value and higher C_2_ parameter value than the virgin SBS/CR modified binder. The fatigue damage should be evaluated in combination with the relationship between the integrity parameter and damage intensity because of the overall effect of the decrease in integrity parameter, which depends on the combined effect of C1 and C2, as presented in Figure 5. Figure 5 shows that the integrity parameter was lost quicker while the thermal oxygen aging was further exacerbated. A significant difference of the loss rate in the integrity parameter with increasing aging conditions could also be found in the three kinds of modified bitumen. The loss rate of integrity parameter of all kinds of modified bitumen before the PAV aging employed a similar increased degree. However, when the aging time of PAV increased to 40 h, the loss rate of the integrity parameter of SBS modified binders was much larger than that of SBS/CR modified binders. The same results happened in the PAV aging for 80 h. The loss rate of the integrity parameter of the SBS/CR modified bitumen, which was aged in PAV for 80 h was nearly to that of the aged in PAV for 20 h. However, it was not be found in SBS-modified bitumen. Table 3 and Figure 5 illustrate that fatigue performance is affected intensely by the C_1_ parameter.

#### 3.2.2. Effect of Aging to Fatigue Life (N_f_) during the Long-Term Aging Processes

The fatigue life (*N_f_*) for all tested binders in the LAS test at strain levels of 1% and 10% are shown in Figure 6. As shown in Figure 6a, the opposite trend of the life cycles with the increasing of aging time at the 1% strain levels was observed in CRMB than that of SBSMB and SBS/CRMB. The value of *N_f_* to fatigue failure of the CRMB became larger when the aging further happened compared with the results shown in Figure 5a. However, the same trend of the life cycles with increasing aging time at the 10% strain levels in all kinds of modified bitumen was consistent with the results shown in Figure 5a. This phenomenon may be led by the crumb rubber powder in CRMB, which was larger than modifiers in the other two modified bitumen. When the strain level was small, the crumb rubber power could strengthen the binders’ fatigue life due to the elastic of crumb rubber. However, when the strain is larger, the crumb rubber power couldn’t endure the deformation in the test, so the trend of CR modified binders’ fatigue life with the increasing aging time became the same as the results shown in Figure 5a.

### 3.3. Low Temperature Performance

Figure 7 and Figure 8 display the effect of aging on the stiffness modulus(S) and creep rate of CRMB, SBSMB and SBS/CRMB in different aging conditions. As shown in Figure 7, in contrast to the stiffness modulus of CRMB and SBS/CRMB, the stiffness modulus of SBS increased sharply after short-term aging, while the other two modified bitumen increased slightly at all test temperatures. The stiffness modulus of the three bitumen increased significantly after PAV aging for 20 h. With the PAV aging time increased, the stiffness modulus of the SBSMB increased faster than that of SBS/CRMB, which indicated that SBS/CRMB held a better ability for anti-aging. Figure 8 revealed that short-term aging had a great effect on the m value of all kinds of bitumen, which was consistent with the result of the stiffness modulus. Moreover, after long-term aging, the creep rate of all kinds of bitumen decreased. With the PAV aging time increased, the creep rate changed similarly to the stiffness modulus of the SBS and SBS/CRMB. However, some limitations still exist when evaluating the low temperature performance of bitumen by S and m values at a certain test temperature. In a number of studies it has been illustrated that it is important to establish a comprehensive indicator that combines m-value and stiffness modulus in the BBR test [15,26,27,28]. Table 4 shows that the SBSMB possessed the same low temperature performance grade (PG) at −24 °C in a low temperature in all aging conditions besides the virgin SBSMB. Therefore, the low service temperature and temperature difference ($\Delta T_{L}=T_{L,s}-T_{L,m}$) was calculated as shown in Table 4.

As shown in Figure 9, when the aging time increased, the PG low temperature decreased in all kinds of modified bitumen. CRMB and SBS/CRMB owned a lower PG low temperature than that of SBS modified bitumen. Moreover, the loss rate of PG low temperature in CR and SBS/CRMBwas similar and smaller than that of SBSMB. It illustrated that CR plays an important role in low-temperature cracking of SBS/CRMB.

As shown in Figure 10, the temperature difference (Δ*T*_L_) of all tested binders was smaller than 0, indicating the binder was more likely to break due to the lack of creep capacity (m-value controlled asphalt). The Δ*T*_L_ values of CRMB in the virgin condition was relatively closer to 0 °C, indicating that compared with the SBSMB and SBS/CRMB, the stiffness and the m-value of CR modified asphalt were more balanced. However, the Δ*T*_L_ values of CR modified asphalt after TFOT and PAV aging dropped sharply, revealing that the aging broke the balance between stiffness and the creep rate. However, for SBSMB, the Δ*T*_L_ values moved closer to zero during the process of aging. Meanwhile, the change of Δ*T*_L_ values of SBS/CR modified asphalt was erratic due to the aging conditions. These results illustrated that the influence of aging in Δ*T*_L_ values of SBSMB was in contrast to that of CRMB.

## 4. Conclusions

In this study, SBS/CRMB was prepared. Low temperature thermal and fatigue crack, permanent deformation performance of CR, SBS and SBS/CR modified asphalt in different aging conditions was also analyzed. The following conclusions were drawn:

(1)The SBS/CR composite modified asphalt possessed the best fatigue resistance, rutting resistance and a low temperature performance before and after different aging conditions. This showed the strong anti-aging ability of SBS/CRMB because of its flexibility and structure that remain in a good condition after long-term aging.(2)Compared with CR and SBS modified asphalt in the virgin condition, the elastic property of CR in SBS/CRMB improved the ability to resist low temperature thermal and fatigue cracking at the range of low and middle temperatures. In the high temperature domain, the copolymer network greatly enhanced the elastic response of the asphalt SBS/CRMB, which shows better deformation recovery.(3)Compared with CRMB and SBSMB under different long-term aging processes, there was a presence of carbon black, which released from the crumb rubber power during the process of the SBS/CR modified bitumen’s preparation. The carbon black could protect the SBS molecule from oxidizing.(4)In contrast to CRMB and SBSMB, it is recommended that SBS/CRMB be used in the subgrade of a high speed railway. It is suggested that in future research, the properties of SBS/CR modified bitumen under different aging times of ultraviolet could be studied.

## Figures and Tables

**Figure 1 materials-12-02345-f001:**
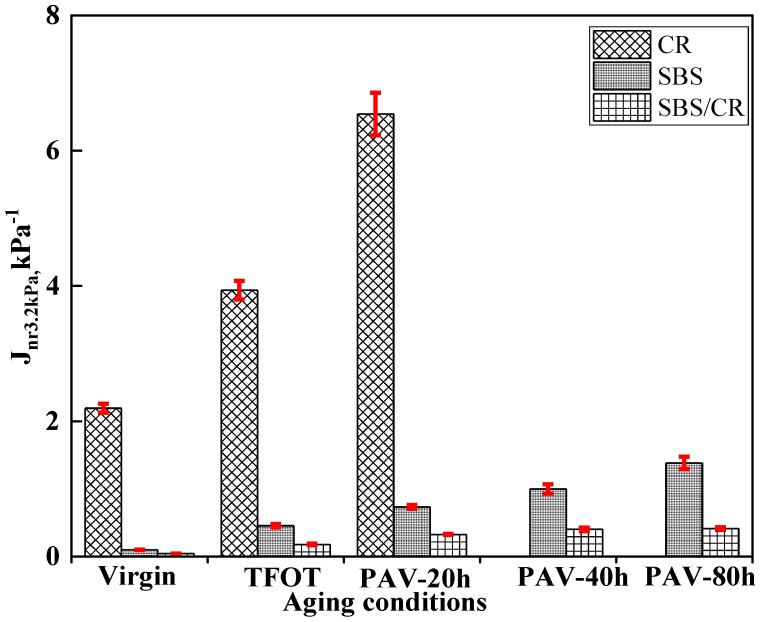
J_nr3.2_ value for three kinds of modified bitumen at different aging conditions.

**Figure 2 materials-12-02345-f002:**
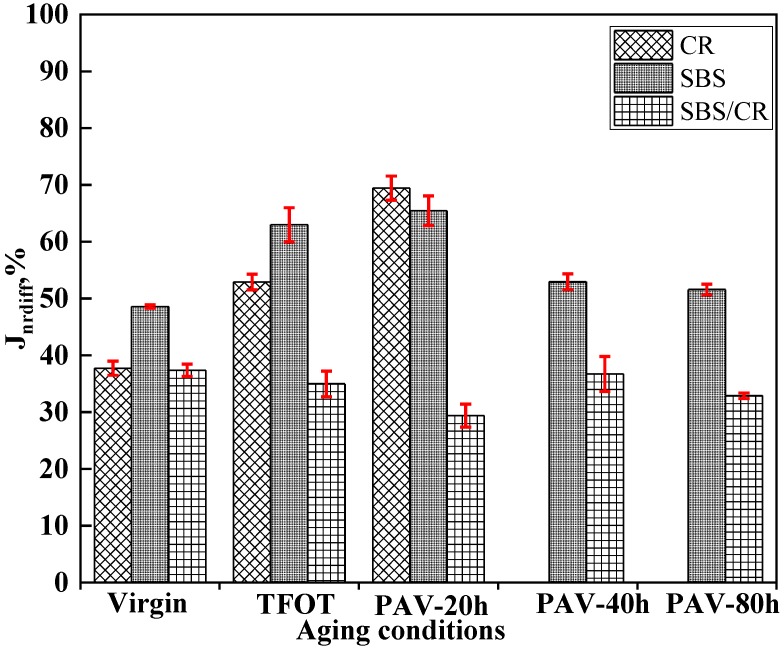
J_nr,diff_ value for three kinds of asphalt at different aging conditions.

**Figure 3 materials-12-02345-f003:**
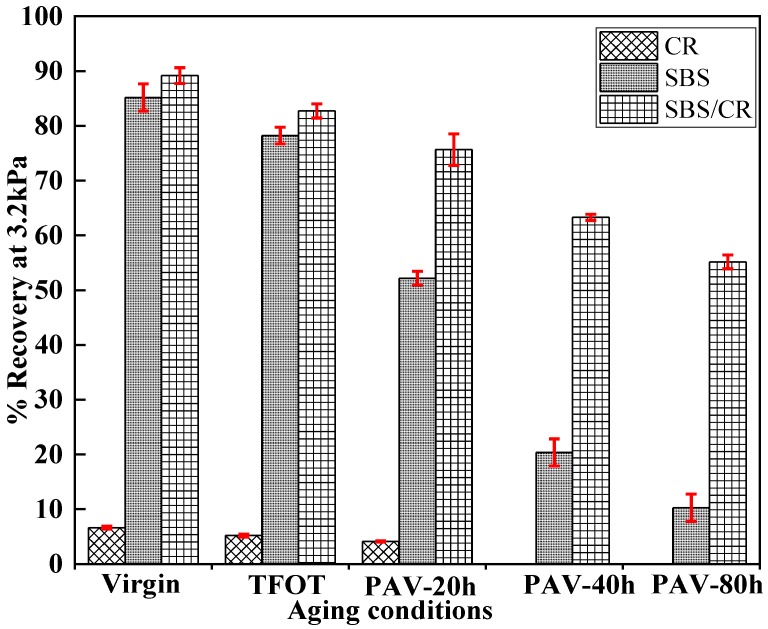
Percent recovery results for all kinds of bitumen at different aging conditions.

**Figure 4 materials-12-02345-f004:**
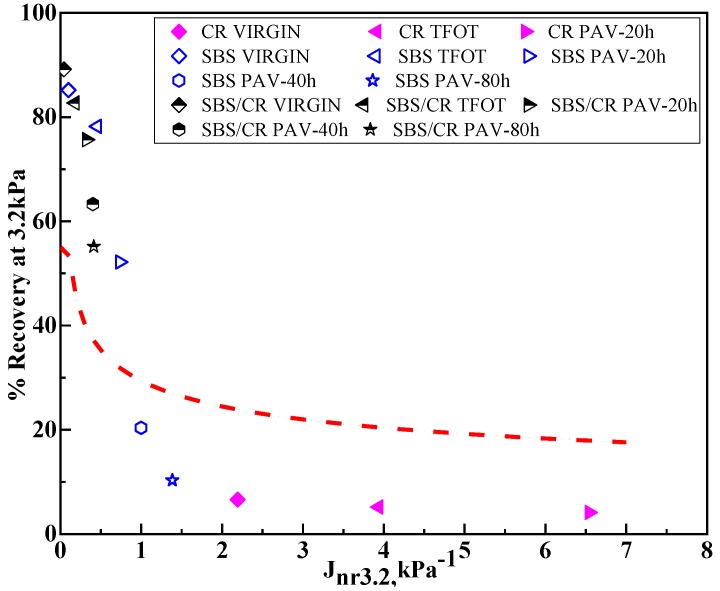
MSCR %R vs. J_nr_ at 3.2 kPa for all tested binders.

**Figure 5 materials-12-02345-f005:**
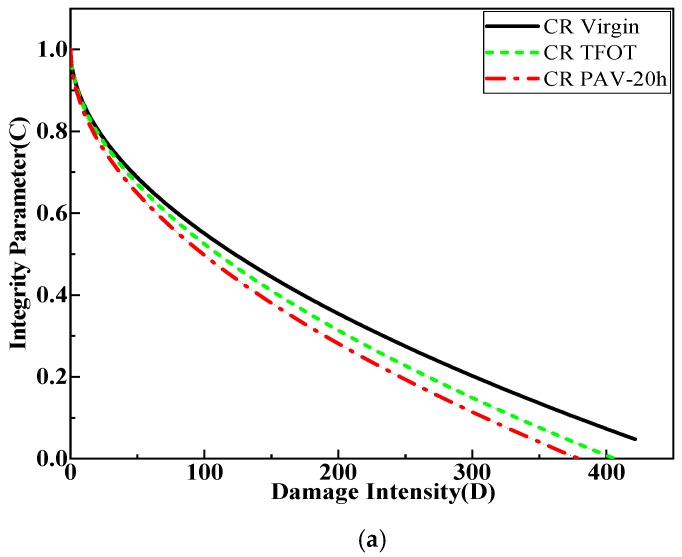
Effect of aging on the curve of integrity parameter vs. damage versus of Crumb Rubber modified bitumen (CRMB), SBS modified bitumen (SBSMB) and SBS/CRMB, (**a**) CR modified binders; (**b**) SBS modified binders; (**c**) SBS/CR composite modified binders.

**Figure 6 materials-12-02345-f006:**
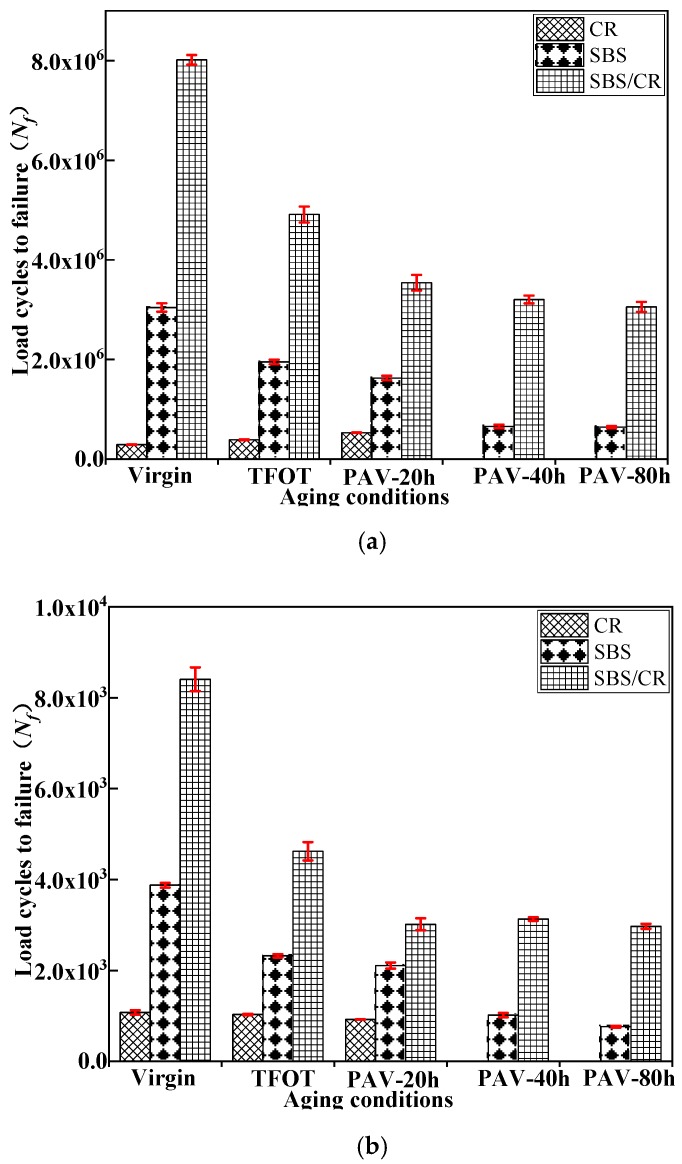
Effect of aging on LAS test fatigue life of CRMB, SBSMB and SBS/CRMB at different strain levels, (**a**) 1% strain level; (**b**) 10% strain level.

**Figure 7 materials-12-02345-f007:**
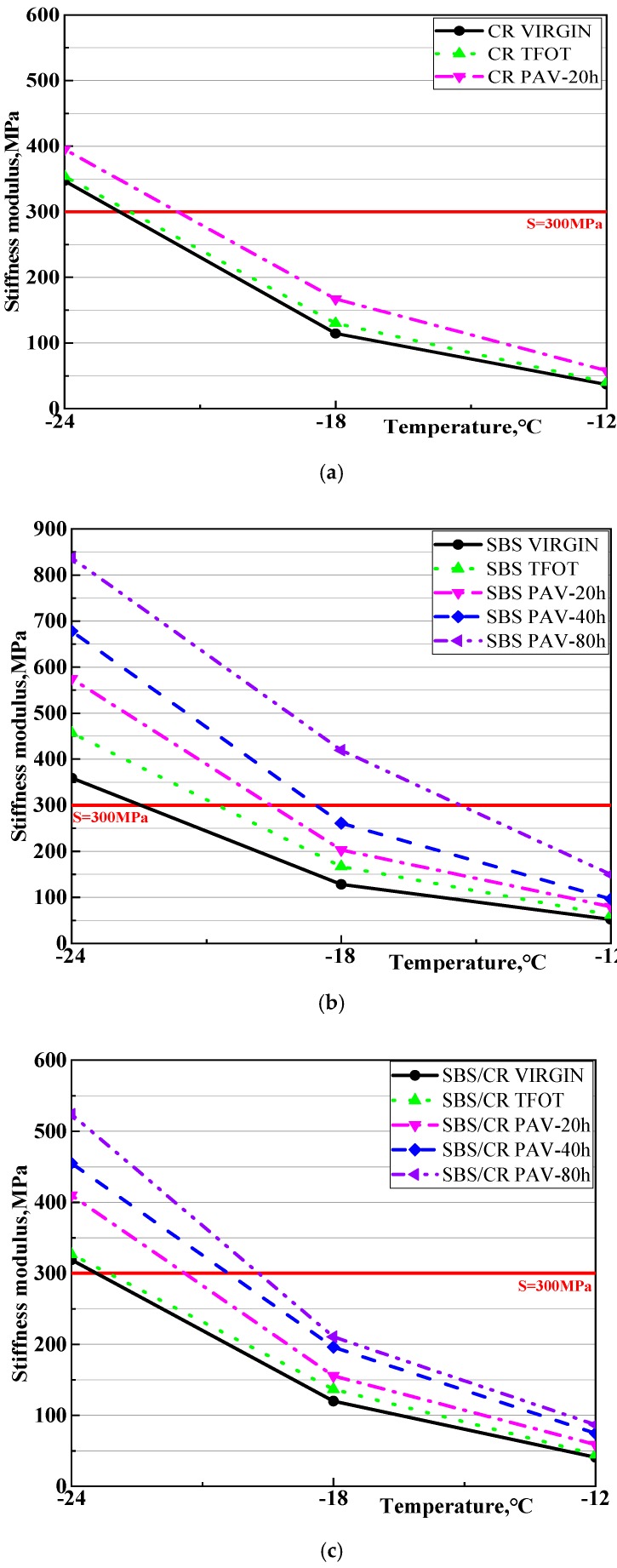
The evolution of stiffness modulus under different aging conditions of all selected asphalt binders, (**a**) Crumb Rubber modified binders; (**b**) Styrene-butadiene-styrene Modified binders; (**c**) Styrene-butadiene-styrene/Crumb Rubber Modified binders

**Figure 8 materials-12-02345-f008:**
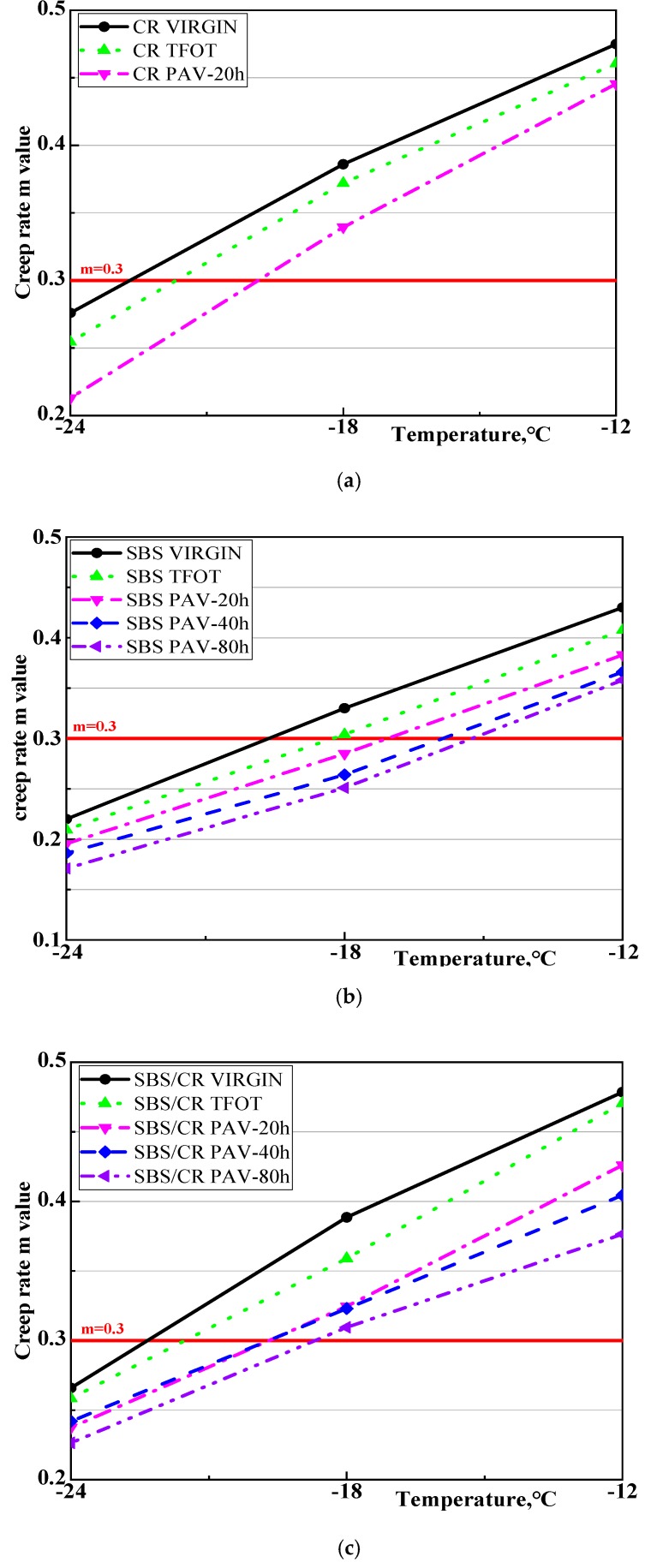
The evolution of creep rate under different aging conditions of all selected asphalt binders, (**a**) Crumb Rubber modified binders; (**b**) Styrene-butadiene-styrene Modified binders; (**c**) Styrene-butadiene-styrene/Crumb Rubber Modified binders.

**Figure 9 materials-12-02345-f009:**
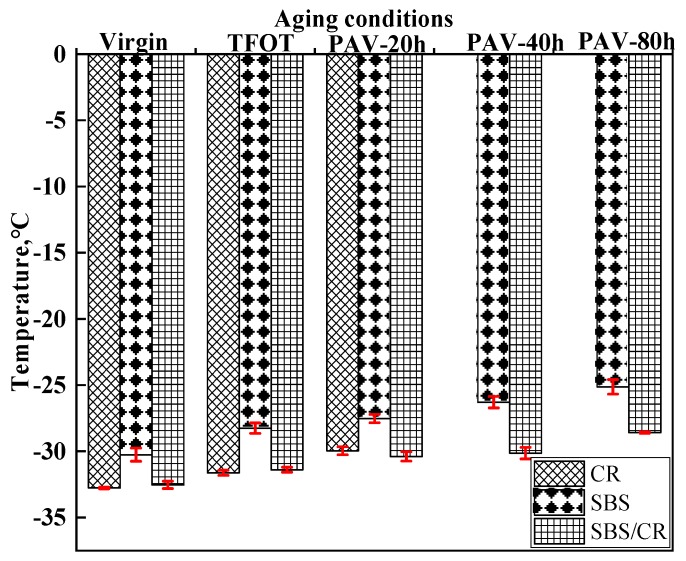
PG Low temperature of all tested binders.

**Figure 10 materials-12-02345-f010:**
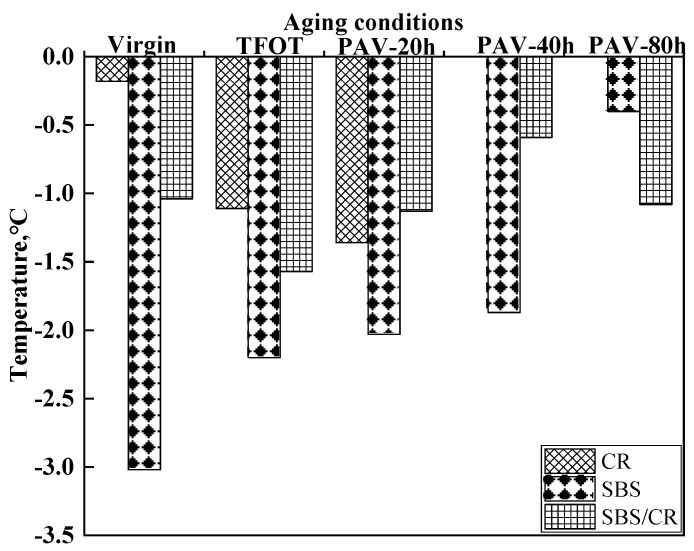
PG Low temperature difference of tested binders.

**Table 1 materials-12-02345-t001:** Properties of base bitumen.

Properties	Unit	Test Results	Test Method
Penetration (25 °C, 100 g, 5 s)	(0.1 mm)	68.9	ASTM D5
softening point (Ring and ball method)	°C	47.2	ASTM D36
Ductility (15 °C, 5 cm/s)	cm	>100	ASTM D113
Change in mass TFOT	%	−0.2	ASTM D2872
Flash point, Cleveland open cup	°C	289	ASTM D92

**Table 2 materials-12-02345-t002:** Multiple Stress Creep Recovery test results for three kinds of asphalt binder in different aging conditions.

Binder Type	Aging Conditions	%Recovery (0.1 kPa)	%Recovery (3.2 kPa)	R_diff_, %	J_nr(0.1 kPa),_ _1/kPa_	J_nr(3.2 kPa),_ _1/kPa_	J_nr,diff,_ %
CRMB	Virgin	23.48	6.63	71.76	1.592	2.192	37.73
	TFOT	21.87	5.19	76.26	2.574	3.936	52.92
	PAV-20 h	17.03	4.13	75.74	3.863	6.542	69.43
SBSMB	Virgin	95.21	85.17	10.54	0.067	0.099	48.59
	TFOT	90.13	78.22	13.21	0.279	0.454	62.97
	PAV-20 h	82.11	52.18	36.45	0.443	0.733	65.45
	PAV-40 h	50.11	20.36	59.36	0.652	0.998	52.93
	PAV-80 h	35.48	10.29	71.00	0.913	1.384	51.57
SBS/CRMB	Virgin	96.01	89.16	7.14	0.031	0.042	37.37
	TFOT	92.71	82.71	10.78	0.132	0.178	34.97
	PAV-20 h	85.22	75.66	11.22	0.251	0.325	29.39
	PAV-40 h	81.21	63.29	22.06	0.294	0.402	36.71
	PAV-80 h	75.71	55.16	27.14	0.310	0.413	32.88

**Table 3 materials-12-02345-t003:** LAS test results of all tested binders based on viscoelastic continuum damage analysis.

Binder Type	Aging Conditions	C_1_	C_2_	A	B	α	$\tau_{\max}$
CR	Virgin	0.041	0.522	291329	2.474	1.237	0.142
	TFOT	0.042	0.530	388559	2.592	1.296	0.162
	PAV-20 h	0.047	0.515	530604	2.760	1.379	0.213
SBS	Virgin	0.050	0.473	3045026	2.894	1.447	0.222
	TFOT	0.047	0.496	1949513	2.888	1.444	0.242
	PAV-20 h	0.056	0.471	1627488	2.922	1.461	0.318
	PAV-40 h	0.058	0.484	657644	2.910	1.455	0.327
	PAV-80 h	0.061	0.536	641767	2.924	1.462	0.324
SBS/CR	Virgin	0.058	0.440	8017435	2.978	1.489	0.1988
	TFOT	0.043	0.504	4911318	3.026	1.513	0.2099
	PAV-20 h	0.042	0.513	3055835	3.042	1.521	0.2496
	PAV-40 h	0.044	0.505	3203419	3.05	1.525	0.2569
	PAV-80 h	0.047	0.498	3542636	3.07	1.535	0.2878

**Table 4 materials-12-02345-t004:** PG Low temperature of BBR test for all tested binders.

Binder Type	Aging Conditions	T_L,m_	T_L,S_	T_L_	ΔT_L(S-m)_
CR	Virgin	−22.77	−22.95	−32.77	−0.18
	TFOT	−2 1.63	−22.74	−31.63	−1.11
	PAV-20h	−19.96	−21.32	−29.96	−1.36
SBS	Virgin	−20.26	−23.06	−30.26	−3.02
	TFOT	−18.26	−20.46	−28.26	−2.20
	PAV-20h	−17.53	−19.56	−27.53	−2.03
	PAV-40h	−16.29	−18.16	−26.29	−1.87
	PAV-80h	−15.13	−15.53	−25.13	−0.40
SBS/CR	Virgin	−22.54	−23.58	−32.54	−1.04
	TFOT	−21.89	−23.46	−31.39	−1.57
	PAV-20 h	−20.39	−21.52	−30.39	−1.13
	PAV-40 h	−20.14	−20.93	−30.14	−0.59
	PAV-80 h	−18.58	−19.26	−28.58	−1.08

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
