# Peer review of "Evolution of Rheological Behaviors of Styrene-Butadiene-Styrene/Crumb Rubber Composite Modified Bitumen after Different Long-Term Aging Processes"

_materials, 2019, doi:10.3390/ma12152345_

Round 1
Reviewer 1 Report
1- What is the storage stability status of the crumb rubber, SBS and the SBS/CR composite modified binders?
2- How did the authors find the 0.2% of sulfur as the optimum content in order to create crosslinking agent?
3-Based on the results shown in Table 4, the CR and SBS/CR modified asphalt binders had almost identical low temperature grade through the SHRP method. The similar results have been reported by other researchers (i.e., Experimental and numerical investigation of low-temperature performance of modified asphalt binders and mixtures; Viscoelastic-based approach to evaluate low temperature performance of asphalt binders) using different approaches. They showed that the SHRP low temperature grading method is not able to distinguish the effect of polymers especially SBS on the low temperature cracking resistance of asphalt binders. The authors may want to review and enhance the introduction by the mentioned references.
Author Response
Point 1: What is the storage stability status of the crumb rubber, SBS and the SBS/CR composite modified binders?
Response 1: The storage stability status of modified binders is a very important parameter, especially for crumb rubber modified bitumen, so in the processes of sample prepared for every rheological test or the aging testing, all kinds of asphalt had been sheared for more than at 500rpm for 15 minutes. In addition, this paper aims to evaluate the rheological properties before and after different long-term aging, so the storage stability status had been ignored.
Point 2:How did the authors find the 0.2% of sulfur as the optimum content in order to create crosslinking agent?
Response 2:In our previous studies(Qinghong Fu, Gang Xu, Xianhua Chen, Jie Zhou,Fengmin Sun. "Rheological properties of SBS/CR-C composite modified asphalt binders in different aging conditions",Construction and Building Materials, 2019 and Rui Wang, Gang Xu, Xianhua Chen, Wenbin Zhou, Hanyu Zhang. "Evaluation of aging resistance for high-performance crumb tire rubber compound modified asphalt", Construction and Building Materials, 2019),it had illustrated form MSCR test frequency sweep test and so on, all test find the sulfur content at the 0.2% is an ideal content.
Point 3: Based on the results shown in Table 4, the CR and SBS/CR modified asphalt binders had almost identical low temperature grade through the SHRP method. The similar results have been reported by other researchers (i.e., Experimental and numerical investigation of low-temperature performance of modified asphalt binders and mixtures; Viscoelastic-based approach to evaluate low temperature performance of asphalt binders) using different approaches. They showed that the SHRP low temperature grading method is not able to distinguish the effect of polymers especially SBS on the low temperature cracking resistance of asphalt binders. The authors may want to review and enhance the introduction by the mentioned references.
Response 3: Thanks for your suggestions, in our revised manuscript, there are more than five references on why we abandoned the low temperature grade to characterize the rheological behaviour of all tested binders.

Reviewer 2 Report
The material evaluated in this study is used for a waterproofing layer under a high-speed rail system. The authors characterized three different compositions of polymer modified asphalt cement for their high, intermediate, and low-temperature performance indices. Those binder properties are popular in the field of asphalt paving where those mechanical properties are deemed desirable to resist certain types of distresses, such as ruttings and crackings. Authors need to discuss the significance of these mechanical properties of asphalt binder as a waterproofing layer on top of a subgrade for a high-speed rail system and show why they thought it's relevant to characterize the material in terms of these properties.
In other words, at the current form, the problem statement is ambiguous, the motivation seems weak, and the experimental design looks simply all-you-can-test.
A more targeted and organized approach is anticipated.
Author Response
Point 1:The material evaluated in this study is used for a waterproofing layer under a high-speed rail system. The authors characterized three different compositions of polymer modified asphalt cement for their high, intermediate, and low-temperature performance indices. Those binder properties are popular in the field of asphalt paving where those mechanical properties are deemed desirable to resist certain types of distresses, such as ruttings and crackings. Authors need to discuss the significance of these mechanical properties of asphalt binder as a waterproofing layer on top of a subgrade for a high-speed rail system and show why they thought it's relevant to characterize the material in terms of these properties.
Response 1: Thanks for your suggestions, in our revised manuscript, we had reorganize the logical relationships. We had introduced the main failure modes of asphalt concrete waterproof sealing layer from our previous studies of theoretical analysis, finite element calculation and past engineering experience. We found low temperature thermal and fatigue cracking, permanent deformation are harmful to the durability of asphalt concrete waterproof sealing layer. Subsequently, we analyzed the effect of aging on the above properties and then explain why we did this research.
